# Thin-Film PVD Coating Metamaterials Exhibiting Similarities to Natural Processes under Extreme Tribological Conditions

**DOI:** 10.3390/nano10091720

**Published:** 2020-08-30

**Authors:** G. S. Fox-Rabinovich, I. S. Gershman, S. Veldhuis

**Affiliations:** 1McMaster Manufacturing Research Institute (MMRI), Department of Mechanical Engineering, McMaster University, Hamilton, ON L8S4L7, Canada; veldhu@mcmaster.ca; 2Joint Stock Company Railway Research Institute, Moscow State Technological University “Stankin” (MSTU “STANKIN”), Moscow 127994, Russia; isgershman@gmail.com

**Keywords:** nanomaterials, self-organization, adaptive nano-structured PVD thin-film coatings

## Abstract

This paper discusses the surface-engineered nanomaterials (adaptive nano-structured physical vapor deposition (PVD) thin-film coatings) that can effectively perform under severely non-equilibrium tribological conditions. The typical features of these nanomaterials are: (a) Dynamically interacting elements present in sufficient amounts to account for its compositional/structural complexity; (b) an initial non-equilibrium state; (c) optimized micro-mechanical characteristics, and (d) intensive adaptation to the external stimuli. These could be considered as functionally graded nanomaterials that consist of two major layers: an underlying (2–3 microns) thin-film PVD coating, the surface on which an outer nanoscale layer of dynamically re-generating tribo-films is produced as a result of self-organization during friction. This tribo-film nanolayer (dissipative structures) was discovered to represent complex matter, which exhibits characteristic properties and functions common to naturally occurring systems. These include adaptive interaction with a severely non-equilibrium environment; formation of compounds such as sapphire, mullite, and garnet, similar to those that arise during metamorphism; ability to evolve with time; as well as complexity and multifunctional, synergistic behavior. Due to several nanoscale effects, this nanolayer is capable of protecting the surface with unprecedented efficiency, enabling extensive control over the performance of the entire surface-engineered system. These surface-engineered nanomaterials can achieve a range (speed and level) of adaptability to the changing environment that is not found in naturally occurring materials. Therefore, these materials could be classified as metamaterials. The second major characteristic of these materials is the structure and properties of the coating layer, which mostly functions as a catalytic medium for tribo-film generation and replenishment. A functioning example of this type of material is represented by an adaptive hard thin-film TiAlCrSiYN/TiAlCrN nano-multilayer PVD coating, which can efficiently work in an extreme environment, typical for the dry machining of hard-to-cut materials.

## 1. Introduction

A top-down approach that mostly focuses on the miniaturization of structural components still predominates in nanoscience and nanotechnology [1]. This miniaturization approach is quite efficient in many fields of science and technology [2].

However, the predominant reductionist paradigm in material science and engineering barely applies to natural systems [3] that are endowed with differing levels of complexity. In light of this problem, a new trend has emerged in surface science, namely nature-inspired surface engineering. Several leading scientific institutions are now moving in the nature-mimicking research direction [4], and principal international conferences have recently taken place on this topic. It has already been established that the investigation of dissipative, non-equilibrium processes, similar to those that can even be observed in the living world, constitute highly promising grounds for future scientific advances [5,6,7,8,9,10,11,12,13]. Therefore, the next step in nanomaterial and nanotechnology development should be the introduction of complex adaptive systems that function in a way designed to imitate natural processes [14]. This nature-mimicking approach should not just be limited to the simple copying of the exact design of natural systems. The time has come to incorporate a temporal approach to the design of such materials [15].

To further proceed in this direction, it is necessary to introduce a somewhat new philosophy based on the paradigm of complexity and the associated theory of self-organization [6,16,17]. Complexity demands a fundamentally different attitude from the mainstream reductionist approach. Although the universal laws are undoubtedly valid, each complex system is different. These differences are of such importance that specific case studies need to be analyzed and understood in each complex system for any conclusion to be carried over to another one [6]. This is why we strongly contend that the approach to the nanomaterial design laid out in this paper would prove useful in a wide variety of applications.

One of the major topics covered in this work is the self-organization phenomenon [18,19,20,21] in a specific area of tribology. A key condition for self-organization is the presence of a strong gradient of characteristics during the non-equilibrium process (for instance, in Benard cells [21]). A common example of such a process is friction [22], which typically exhibits a strong gradient of various characteristics (for example, temperatures and stresses) among the surfaces in tribo-contact [22]. This presents an environment under which potential self-organization phenomena may readily emerge to their full extent [22].

Currently, the intense demand to improve the efficiency of various engineering processes results in increasingly severe and even extreme operating environments [23]. In the present paper, we use extreme tribological conditions as such an example. In this case, tribo-systems can almost always be found to operate under strongly non-equilibrium conditions [24].

One significant example of such conditions is manifested under the ultra-performance dry (lubricant-free) machining (in particular, ball nose end milling) of hard-to-cut materials when the tribo-system operates under heavy (up to 3–5 GPa) loads and high (1000–1200 °C and above) temperatures [25]. Such severe cutting conditions can be withstood by cutting tools with adaptive nanostructured thin-film wear-resistant physical vapor deposited (PVD) coatings [23,26]. A major feature of such coatings is their ability to form highly protective/lubricating tribo-films on the friction surface as a result of the interaction with the external environment [24,27]. The application of these adaptive coatings can lead to an unprecedented increase in machining productivity [28].

A new generation of wear-resistant thin-film coatings is introduced in this work, which represents a specific example of a complex adaptive system. Such systems are capable of fully developing their unique properties during temporal interaction with the ever-changing, severe environment they are embedded in [29,30,31,32,33,34]. The probability of self-organization’s incidence is higher in complex systems [35]. Therefore, complex systems, contrary to expectation, can spontaneously exhibit stunning degrees of order, which, in turn, is essential for the understanding of their emergent (synergistic) behavior [36,37,38]. A significant recent achievement in surface engineering was the introduction of coatings with a complex composition, which contain at least five alloying elements. This was accomplished in two different ways: High-entropy alloy coatings [39,40,41,42,43,44,45] and coatings with emergent performance [27].

The coating with emergent properties [27] showed much promise under severe external conditions associated with the machining of hard-to-cut materials. The next step to improve upon was the coating’s architecture: multilayered and bi-multilayered [28]. Combining these two approaches (compositional and structural optimization) resulted in the development of efficiently operating complex adaptive systems.

The goal of this paper is to demonstrate a complex adaptive system using up-to-date results from the gradual compositional and architectural optimization of a surface-engineered layer, which is represented by a bi-nano-multilayer hard thin-film TiAlCrSiYN/TiAlCrN coating deposited by PVD.

## 2. Adaptive Surface-Engineered Systems

The adaptive of hard thin-film PVD coatings [46], as well as self-lubricating hard coatings [47], had been developed more than a decade ago [48]. The adaptive features progress in time through the permanent regeneration of surface nanoscale tribo-films caused by the interaction of the coating material with the external environment [49]. This is directly related to the process of self-organization during friction [48].

### 2.1. Tribofilms Characteristics: Composition, Structural Characteristics, and Temporal Behavior of Tribo-Films

These surface films are generated within the system as a consequence of intensive matter and energy flow, i.e., irreversible processes. Such structures, termed as “dissipative structures” by Prigogine [20,21], spontaneously form under strongly non-equilibrium conditions and could be regarded as a new type of dynamic material state. Dissipative structures can be distinguished from equilibrium structures because they are, at the core, processes. The origins of these structures are intense non-spontaneous processes with negative entropy production that previously did not exist in the system. The driving force behind self-organization is the tendency of open systems to protect and stabilize themselves, as evidenced by the decrease in entropy production during non-equilibrium processes [48]. Although dissipative structures are a process, they still remain on the surface after friction is stopped. As meta-stable structures, they are critically important process indicators and have been studied in detail [35].

Tribo-films are an integral part of modern adaptive surface-engineered nanomaterials. Investigation of meta-stable tribo-films provides crucial information about the characteristics of dissipative structures generated during friction. Furthermore, these characteristics can be manipulated to ensure a superior wear performance for the entire tribo-system. Tribo-films are the major integrated part of modern adaptive surface-engineered nanomaterials. As dissipative structures generated during self-organization, tribo-films are dynamic surface spatial structures with temporal behavior. As tribo-films form in situ during friction, they became originally known as ‘secondary structures’ [48]. Such structures determine the adaptive behavior of the studied complex adaptive tribo-system. During friction, these tribo-films are produced on the base surface of the engineered material through structural modification and interaction with the environment (mostly with surrounding oxygen, Figure 1) [48,49,50,51,52,53,54,55,56,57].

Figure 1 shows the scanning Auger image of the worn surface of the cutting tool with a multilayer TiAlCrSiYN/TiAlCrN PVD coating [11]. The general view of the worn area is shown in Figure 1a. A detailed distribution pattern of aluminum-based tribo-oxides and non-stoichiometric nitrides within the worn area is depicted in Figure 1b. The resultant Auger image illustrates the meta-stable phase distribution within the worn area. Using Auger imaging, we can show “snapshots” of the dynamically regenerating surface films caused by a series of tribo-chemical reactions on the friction surface (Figure 1).

#### 2.1.1. Composition of the Tribo-Films

Published X-ray photoelectron spectroscopy (XPS), as well as transmission electron microscopy (TEM), data indicate the formation of elementary Ti, Al, Cr, Si, and Y tribo-oxides on the friction surface of the TiAlCrSiYN/TiAlCrN coating [12,13,49]. All of these ‘primary’ tribo-oxides interact with each other during wear, further forming complex oxide films such as Al_6_Si_2_O_13_ mullite [58,59], garnets [59], and even ruby Al-Cr-O tribo-phases [60,61] (see Figure 2).

These complex oxides have significantly lower thermal conductivity compared to the major primary tribo-phase (sapphire) [60,61]. The formation of complex oxides significantly improves surface protection. Such tribo-oxides are produced by a non-spontaneous, nonlinear process, which decreases entropy production in the system during the initial running-in stage of wear, in a manner that could be seen as being somewhat analogous to an autocatalytic reaction [62,63].

The temporal process outlined above (i.e., formation of complex matter [5]) is also a dissipative structure that corresponds to negative entropy production and requires a lot of energy to commence. Consequently, less energy is left in the system for surface damaging processes. In tribo-systems with growing complexity, the number of initiated processes corresponds to the amount of interactions. The nonlinear process described above is associated with the formation of a new dissipative structure. In this way, the wear rate could be additionally reduced.

The main topic of interest in this paper, however, concerns the relation between processes occurring under extreme tribological conditions at the nanoscale, to certain naturally occurring processes. Contemporary industrial machining frequently involves extreme frictional conditions, such as high temperatures between 900 and 1300 °C and pressures of up to 5 GPa [25]. Such conditions represent a specific example of an extreme external environment. Moreover, these conditions are strongly non-equilibrium, due to a very strong gradient of properties typical for friction surfaces [22].

This process is especially intensive during the running-in stage when the coating layer exhibits the most intensive adaptive response to severe external stimuli [12,23,35]. Our observations have demonstrated that under the outlined conditions, processes on the frictional surface occur on a nanoscopic scale and over a very brief period of time (a few minutes) [35]. As it was outlined above, under extreme conditions, various tribo-phases have formed on the friction surface, such as sapphires, mullites, garnets, and other high-temperature thermal barrier and lubricating compounds [35]. A remarkable analogy can be drawn between the formation of these tribo-phases and the very-long-timescale (1–5 million of years) ultra-high-pressure metamorphic processes that occur within the mantle of the Earth. The reason why this is mentioned here is that metamorphism has a direct connection with the process of self-organization of dynamic systems studied in geoscience [49,50]. For example, diamond crystallization occurs under somewhat similar temperature/pressure conditions: ~1100 °C and P ~7–8 GPa [64]. Recently discovered non-kimberlite diamond-bearing rocks contain micro-diamonds of nanometric sizes [65]. They arise as inclusions in garnet and zircon: Some are situated at the grain boundaries of phengite, quartz, garnet, and sapphire. Some inclusions have an Al-Si-O composition [65] and could be amorphous as well as crystalline. The extraordinary aspect of self-organization processes in surface-engineered systems operating under extreme frictional conditions is that all of the aforementioned compounds that exist within the Earth’s mantle [66,67] can also be found on the surface of the cutting tool after a very brief period of time (a few minutes of operation) within the initial wear stage [49].

Therefore, it is quite plausible that a connection exists between highly non-equilibrium tectonic changes occurring over eons in nature and rapid nanoscale surface effects that develop under extreme frictional conditions. We have to take into consideration that the frictional systems considered also work under a similar range of ‘mega’ stresses and temperatures, as well as far from equilibrium external stimuli [49].

Speaking of surface phase transformations, the major obstacle to achieving and activating beneficial self-organizing processes in general scientific practice is that many engineering systems simply have never reached such an extreme state of non-equilibrium. Therefore, a system is specified in this study in which the products are not just static assemblages of components associated with self-assembly but are distinct, newly functioning entities. In this case, these are nanoscale dissipative structures (surface tribo-films), which, due to the adaptive response of the tribo-system to harsh environments, have unprecedented protective properties under extreme operating conditions [49].

#### 2.1.2. Structural Characteristics of Tribo-Films in Relation to Their Spatial/Temporal Behavior

A number of nanoscale effects are associated with the temporal behavior of the tribo-film layer and strongly affect the progression of wear. The wear vs. length of cut data for cutting tools on which the outlined coating was applied is presented in Figure 3.

The wear rapidly grows during the initial running-in stage, before stabilizing and transforming into the stable post running-in stage (Figure 3), which, in turn, eventually culminates in the catastrophic wear stage. The running-in period is when self-organization actually commences and then progresses into the post-running-in (stable stage) as the wear rate stabilizes [35]. The tribo-films’ temporal behaviour is strongly concurrent with the progression of wear. At the very beginning of wear, when chip formation gives rise to intensive thermo-mechanical processes (adhesive interaction at the tool/chip interface, heating, frictional load), it becomes associated with the stick–slip phenomenon, which is directly related to the self-organized critical process (Figure 3b) [49,68,69]. This, in turn, results in the growth of entropy production [35], which immediately prompts an adaptive response from the highly non-equilibrium surface-engineered layer to external stimuli and directly leads to intensive processes with positive entropy production, i.e., to the intensification of energy dissipation (self-organization) [8,35]. Secondary structures emerge during the initial stage of friction at a highly rapid rate (Figure 4).

XPS data (Figure 4) demonstrate an increased amount of protective tribo-ceramic films emerging at the very beginning of the wear process. It has to be noted that the crystal structure of these tribo-films is amorphous at this stage of wear, which promotes energy dissipation (see Figure 6 for more detail).

The amount of oxide tribo-films is minimal (below 1–2%) prior to cutting. The greatest total amount of tribo-ceramics (around 36.1%) forms only after 1 min of cutting (at a length of cut of 2 m, which is less than 1% of the entire cutting period before the end of the tool’s lifetime) (Figure 3). During further stages of wear, the amount of tribo-films is lower (around 28 at.%) and remains stable with wear time (Figure 4). This is consistent with literature results that state that the greatest probability for self-organization to occur in this tribo-system is during the initial stage of cutting [11]. Once friction is stabilized, entropy production will remain very low in the post running-in stage, (after a length of cut of over ~30 m, Figure 3), due to permanent tribo-film replenishment [11]. The friction surface is thereby sustainably protected by the formation of a constant amount of tribo-films on it (Figure 3). This phenomenon can be termed as a trigger effect.

The trigger effect is a feature of surface-engineered systems specially designed to stabilize wear rate during the shortest period of time and possibly lowest process intensity. The trigger effect can be observed in surface-engineered materials working under high-temperature and high-stress conditions. An optimized composition, a special initial structure of the surface layer (see Section 2.2), strongly enhances this effect and leads to the significant improvement in tool life.

Direct TEM observations show that the layer of dynamically replenishing tribo-films lies in the range of nanometers (thickness of around 6 nm) (Figure 5). This is confirmed by similar data obtained by an AES depth profile [70,71].

Tribo-films have a very complex amorphous/crystalline microstructure [11] that evolves with time. The tribo-films observed under extreme tribological conditions are a mixture of two types of tribo-films (amorphous and crystalline) known from general tribology [50,57], which is confirmed by the Fourier transforms presented in Figure 6.

Amorphous-like tribo-films may exhibit super-plasticity, thus promoting energy dissipation during friction [55]. In addition, amorphous-like films possess a high thermal protective ability [55]. The other identified phases are crystalline tribo-ceramics. Depending on the chemical composition, tribo-films of this type feature either lubricating properties (such as high-temperature lubricating oxides [46]) or surface-protective properties due to their thermal barrier, high chemical and thermodynamic stability, and high hot hardness [53,56,57]). It is shown in Figure 1 that separate islands of non-equilibrium nitrides and oxides are formed during the initial stage of wear. At the beginning of the running-in stage (cutting length of 2 m), the oxides had an amorphous structure, according to electron energy loss fine structure (EELFS) data (Figure 6a) [11].

Only the nearest coordination spheres with radii of up to 0.4 nm form the peaks. Figure 6b shows that after a length of cut of 15 m (within running-in stage), a mixture of Al and Si oxides with a mullite-like structure is formed. The intensity of the F(R) characteristic is slightly increased. These details are indicated by the blue ovals in Figure 6. After a length of cut of 15 m, nano-crystalline/amorphous oxides form on the friction surface. At a length of cut of 30 m, an increased variety of tribo-oxides is produced on the worn surface (Figure 6c,d). Sapphire-like Al_2_O_3_ and Cr_2_O_3_ tribo-oxides with a large amount of structural defects were also observed. The sapphire-like tribo-films were observed after 60 m of cut (Figure 6d). These tribo-films are characterized by a stronger long-range order in the 0.4–0.8 nm region of the nearest atomic surrounding, compared to those of mullite and chromium tribo-oxides formed at a 30 m length of cut (Figure 6c).

At the end of the running-in stage and during the entire stable wear stage, tribo-oxides cover the tool surface. The wear rate of the oxide films is very low due to their exceptional protective properties [58,59,60] under high temperatures that develop during outlined extreme tribological conditions. This leads to recrystallization processes within the layer of the oxides [70].

However, once the wear process develops in time (within the steady stage of wear), the atomic structure of the tool surface changes again (see Figure 6e,f, after 100 m length of cut): At a small (less than 1 nm) electron beam size, the EELFS reveals areas with an amorphous structure similar to that presented in Figure 6a. The long-range order of the tribo-films disappears and their structure becomes amorphous again.

As mentioned in [13,56], these amorphous zones form due to the accumulation of damage in the upper layer of the nitride coating and localized detachment of the “old” oxides. Fresh amorphous oxides form on the open surface of the nitride coating layer in a similar way as during their tribo-oxidation at the beginning of the cut (Figure 6a,f).

Another nanoscale spatial effect could also be described. According to the principle of ‘dissipative heterogeneity’ [54], a greater part of the interactions between frictional bodies is concentrated within the thin layer of the tribo-films [23]. The depth of such a layer is lower by an order of magnitude than that which is typically associated with wear and surface damage phenomena [54].

The phenomena involved in this process determine the performance of the tribo-films. Friction periodically generates a very intensive heat flux at the surface. This heat flux is mostly transferred to the external environment due to the very low heat conductivity of the tribo-oxides nano-layer [70]. Most of the thermal protection can be attributed to the mean free path of phonons within the tribo-oxides nano-layer reaching or exceeding the thickness of the tribo-oxide films [72]. This blocks thermal transport through the tribo-film layer. The scattering of phonon oscillations on account of numerous defects and interfaces is another reason for the loss in heat transfer through the tribo-films [73]. Another contributing factor is the high density of lattice defects generated by high pressure on the friction surface under extreme tribological conditions [74]. All of these factors work synergistically to result in enormously high thermal barrier properties that develop within the nanoscale tribo-film layer [75]. Therefore, for the most part, heat is rendered incapable of being transferred into the body of the coating layer. Instead, it dissipates via chip removal and radiation to the surrounding environment.

The ability of the tribo-film layer to strongly promote heat dissipation accounts for the dramatic temperature gradient observed within the nanoscale tribo-films layer [13,56]. The proposed mechanism of heat dissipation at the nano-scale prevents damage to the tribo-film/coating interface with unprecedented efficiency and consistency [75]. As a result, a very strong temperature gradient was achieved under extreme operational conditions, significantly reducing the temperature beneath the thermal barrier/lubricating nanolayer of the tribo-films, as was clearly shown by combined TEM and synchrotron radiation technique, such as the X-ray absorption near-edge structure (XANES) method [56], studies of the worn coating layer [75]. Below the nanoscale tribo-films layer temperature drops by a few hundred degrees C (estimated at around 400 °C) [61], providing excellent protection of the friction surfaces and overall performance control of the entire surface-engineered system [75]. This observed phenomenon can be contingently termed as a ‘heat flow reflection effect,’ which involves many simultaneously developing processes outlined above and is strongly associated with synergistic, emergent-like behavior of complex matter [5,16]. These two surface effects (trigger effect and heat flow reflection effect) result in efficient protection of the underlying coating layer under operation and convert the entire surface-engineered tribo-system into a functionally graded nanomaterial.

In the nano-multilayer TiAlCrSiYN/TiAlCrN coating, various tribo-films form on the friction surface, with each performing a specific task (Figure 1 and Figure 2). However, the films responsible for wear resistance under extreme operating conditions are thermal barrier tribo-ceramics with a sapphire/mullite structure that effectively protects the friction surface at the very high temperatures typical under the outlined conditions. The mullites (as well as sapphires) are known to be the hardest of the oxide crystals, possessing a high strength at high temperatures [71,72], excellent capability to absorb the energy of external impact, and excellent thermal shock resistance at high temperatures [71,72]. Although mullite has a lower thermal conductivity [72,73], the sapphire provides superior fracture resistance [73,74]. In the sapphire/mullite tribo-film, the mullite’s brittleness is compensated by the more fracture-resistant and thermally conductive sapphire. We can assume that the mullite tribo-phase is mostly responsible for the accumulation of frictional energy. At the same time, the less hard and more fracture-resistant sapphire also promotes energy dissipation.

Moreover, as the surface-engineered system strives to be sustained in its environment, it is capable of mobilizing its entire potential. This could be illustrated by the ability of the system to form an Y_2_O_3_ tribo-oxide. The coating layer contains only a small amount of Y (2 at.%). Nonetheless, this element is intensively transmitted to the surface, forming a very beneficial high-temperature protective/lubricating oxide tribo-ceramic [75] (Figure 7). A similar trend can be observed with Si [75].

In general, functionally graded materials belong to a class of advanced materials with varying properties over a changing dimension [76,77]. In the case of a coating, this dimension is the thickness of the surface-engineered layer [78]. It is important to note that such materials occur in nature [79]. The functionally graded material replaces the sharp coating/substrate interface due to the lower hardness of the sublayer [28] with a gradient interface that ensures the smooth transition of the surface-engineered material to the substrate and postpones failure of the coating layer (Table 1) [28,79,80].

One unique characteristic of functionally graded materials is their ability to be tailored for a specific application [76,78,79], especially under intensifying tribological conditions. The functionally graded nanomaterials outlined in this study are highly ordered systems, dynamically embedded within an increasingly severe environment. They possess high adaptive capacity and, as such, become capable of sustaining these extreme operating conditions with unattainable levels of tribological performance. As external conditions become more severe, entropy production within the surface engineering system is expected to increase, which leads to a higher wear rate. However, the results presented in Figure 8 show that the increasing severity of the operating conditions (in this case, various cutting speeds) produces the reverse effect—the wear rate actually reduces. Figure 8 shows the growth in wear resistance after the cutting speed is increased by 2.5 times caused by the formation of a greater amount of protective tribo-films during these intensified conditions (sapphire + mullite at a higher speed vs. gamma-alumina at the lower speed) [58,59]. This means a more severe environment actually raises the adaptive capacity of complex adaptive engineering systems [75], due to the formation of tribo-films with enhanced protective/lubricating capabilities. As it follows from the data presented in Figure 8, the presented nanomaterial can reach a degree (speed and level) of adaptability to the changing environment that is not found in natural materials. Therefore, the nanomaterials outlined in this study could be considered as meta-materials.

Such materials are artificially designed to mostly consist of nanostructures engineered to possess properties not found in naturally occurring materials [80]. These materials have the ability to respond to external stimuli in new ways [81]. In the cases outlined, this is achieved through the enhanced adaptive response of the surface-engineered layer to the increasingly severe environment.

### 2.2. Coating Layer

The second component of the surface-engineered nanomaterial is the several-micron-thick thick layer of a PVD TiAlCrSiYN/TiAlCrN coating. This layer functions as a medium for the catalysts of tribo-film generation and replenishment. On its own, this layer also represents a complex system, whose composition, architecture, structure, and properties are carefully tuned [27,28,80,81].

This 2–3 micron-thick hard PVD coating has the following outstanding characteristics: High compositional complexity, a bi-multilayer architecture, a nano-crystalline/laminated structure, and a strongly non-equilibrium state due to the high amount of defects in its crystal structure in the initial state [12].

The structure of the bi-multilayer TiAlCrSiYN-based thin-film coatings (approximately 2–3 microns thick) is presented in Figure 9.

The multilayer coating has a complex structure that combines a modularly composed nano-multilayer structure (nano-layer period of 20–40 nm) with a columnar nanostructure (Figure 9a) [28]. The nano-layers perfectly align with the shape of the sharp cutting edge (Figure 9b). There is no visible damage within the entire coating layer despite significant (close to 90°) bending of the nano-layers. It was previously demonstrated [31] that the bending stresses of the individual nano-layers in a multilayer coating are lower due to the accommodation of the bending radius that causes fracture in a monolayer coating of similar total thickness. Highly compressive residual stresses also contribute to this effect. However, this is quite remarkable for the layer with a hardness above 30 GPa (Table 1). All these features also contribute to the sustainability of the coating layer under operation, strongly improving wear performance. As was outlined above, the studied surface-engineered layer (bi-multilayer coating) possesses significantly better scratch crack propagation resistance (CPRs parameter) compared to the multilayer coating (Table 1). This is most likely due to the presence of a slightly softer TiAlCrN interlayer [28], which improved the adhesion of the bi-multilayer coating to the carbide substrate. Failure of the coating layer is thus postponed, which is a typical outcome achieved by the functionally graded surface-engineered material.

## 3. Conclusions

This paper outlines the potential benefits to nanoscience that can be provided by the study of phenomena occurring under harsh environmental conditions. A high compositional complexity in combination with nano-structural characteristics may result in the development of a surface-engineered nanomaterial with unprecedented ability to sustain these conditions. This approach could be effectively applied in a variety of challenging practical fields. A novel generation of adaptive nano-structured PVD thin-film coatings and, in particular, the TiAlCrSiYN/TiAlCrN bi-multilayer PVD coating is introduced in this work, which could successfully function under strongly non-equilibrium tribological conditions combined with high temperatures (about 1000–1200 °C) and stresses (up to 5 GPa).

The surface-engineered nanomaterial considered in this paper has some characteristics typical of complex adaptive systems: A large number of dynamically interacting elements that result in compositional/structural and architectural complexity, an initial non-equilibrium state of the coating layer, openness of the tribo-system via adaptation to a severely nonequilibrium environment, and, lastly, optimized mechanical characteristics.

These surface-engineered layers could be considered as functionally graded nanomaterials that consist of two major, differently scaled layers: (1) The outer, 2–5 nm-thick layer of dynamically regenerating tribo-films that form as a result of interaction with the environment. This layer is produced as a consequence of self-organization during friction with the formation of dissipative structures on top of (2) the underlying 2–3 micron-thick PVD coatings layer.

The dynamically formed top nano-layer of the tribo-films represents complex matter. As such, it exhibits the following characteristics and functions common to natural systems:openness by means of adaptive interaction with a non-equilibrium severe environment,formation of the compounds such as sapphire, mullite, garnet, and others, similar to those that develop as a result of metamorphism,temporal behavior and ability to evolve with time,complexity and multifunctionality,emergent (synergetic) performance.

The tribo-film nanolayer can efficiently protect the frictional surface, thereby enabling control over the performance of the entire coating layer. Two recently discovered nanoscale phenomena are described: The “trigger’’ effect and the ‘’heat flow reflection’’ effect. Due to these effects, the surface-engineered nanomaterials can reach a range of adaptability to the increasingly severe environment that is not found in natural materials. Therefore, the nanomaterials outlined in this study could be considered as metamaterials.

The second functionally graded nanomaterial layer discussed here, i.e., the micron-thick coating layer, mostly serves as a medium for the catalysts of tribo-film regeneration. To promote this function, its architecture, structure, and properties have to be carefully optimized.

As the presented nanomaterials are highly ordered adaptive systems, dynamically interacting with an increasingly severe environment, they are capable of effectively sustaining extreme operating conditions with unattainable levels of adaptability and wear performance. A true understanding of such metamaterials can only be obtained by a holistic approach.

## Figures and Tables

**Figure 1 nanomaterials-10-01720-f001:**
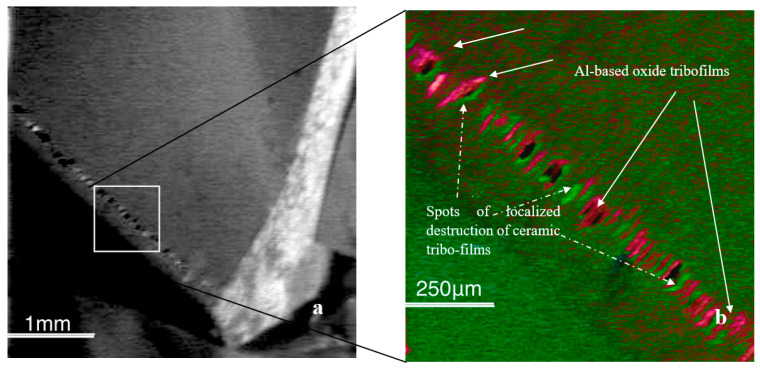
Auger image of atomic-scale tribo-films forming on worn surface (**a**)—a general image (topography contrast); (**b**)—red image, tribo-films; light green image, coating layer [11].

**Figure 2 nanomaterials-10-01720-f002:**
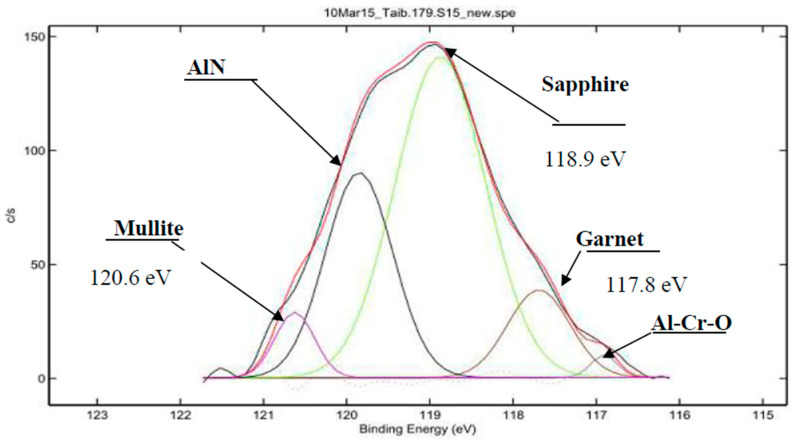
HR XPS spectra of complex tribo-oxides forming the beginning of the running-in stage (15 m length of cut): Al 2s spectrum [49].

**Figure 3 nanomaterials-10-01720-f003:**
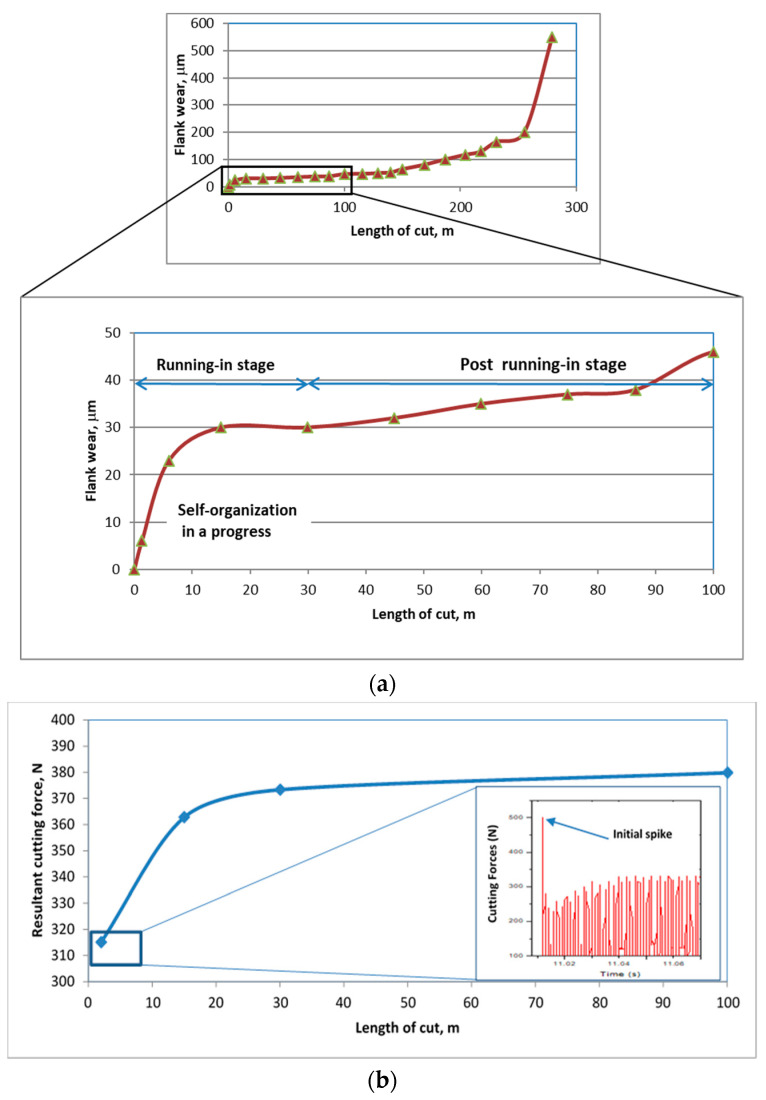
Flank wear vs. length of cut data for ball nose end mills with TiAlCrSiYN/TiAlCrN multilayer coating: (**a**) Wear curve with an indication of specific stages of wear: Running-in and post-running-in (stable) stage of wear; (**b**) cutting forces data within first 3 m length of cut [49].

**Figure 4 nanomaterials-10-01720-f004:**
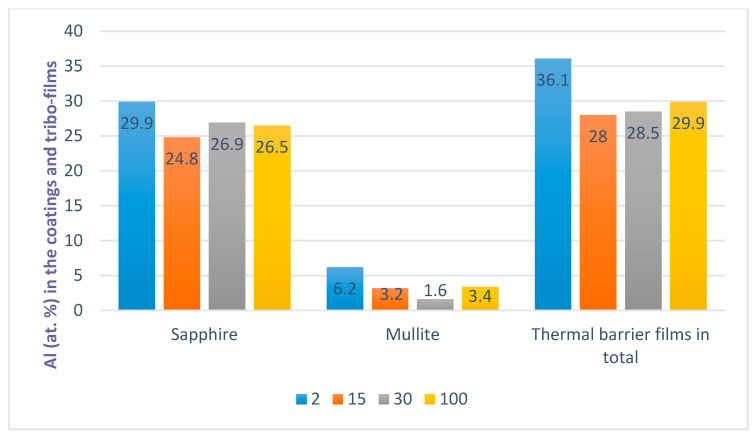
XPS data on the tribo-films formation vs. length of cut. Amount of tribo-films vs. length of cut: 2; 15; 30; 100 m [49].

**Figure 5 nanomaterials-10-01720-f005:**
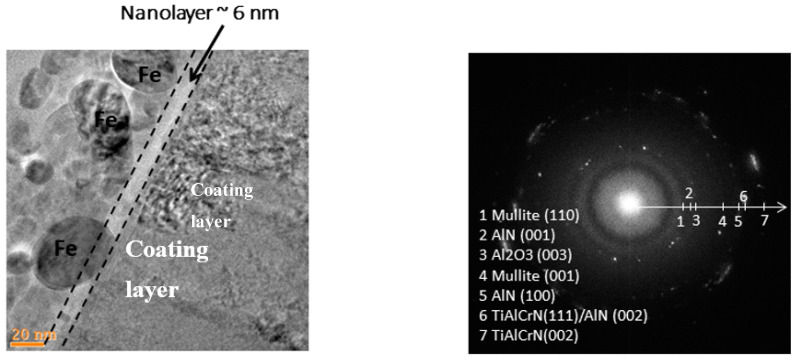
High resolution TEM image of the Focussed ion beam (FIB) cross-section with selected area electron diffraction (SAED) pattern insert of the worn surface, 15 m length of cut: Nano-layer of tribo-films on the surface of TiAlCrSiYN/TiAlCrN coating (6 nm thick) [49].

**Figure 6 nanomaterials-10-01720-f006:**
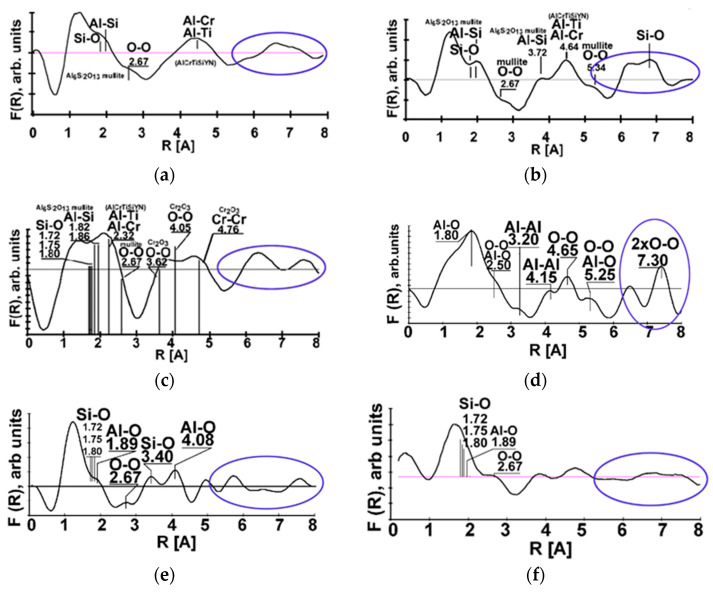
The evolution of the atomic coordination R [Å] in the tribo-oxides vs. cutting length: (**a**) 2 m; (**b**) 15 m. The blue oval indicates the long-range order region (electron energy loss fine structure (EELFS) Fourier transforms); (**c**) 30 m, mullite-like tribo-oxides; (**d**) 60 m, sapphire-like tribo-oxides; (**e**) 100 m, mullite-like and amorphous nano-layer, and (**f**) after length of cut of 100 m [11].

**Figure 7 nanomaterials-10-01720-f007:**
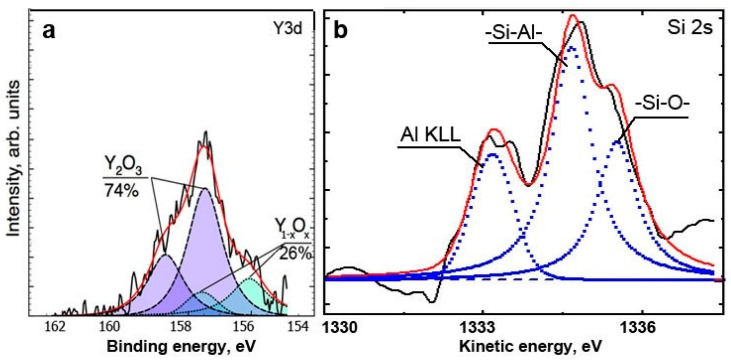
HR XPS Y3d (**a**) and Si2s (**b**) photoelectron spectra of worn surface of multilayer TiAlCrSiYN/TiAlCrN coating—within running-in stage of wear (after length of cut of 15 m) [11].

**Figure 8 nanomaterials-10-01720-f008:**
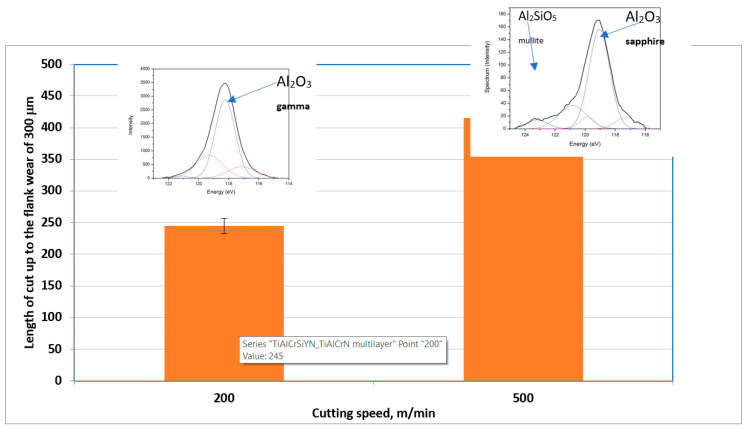
Improvement in wear performance with severity of tribological conditions in relation to the tribo-films formation on the friction surface [75].

**Figure 9 nanomaterials-10-01720-f009:**
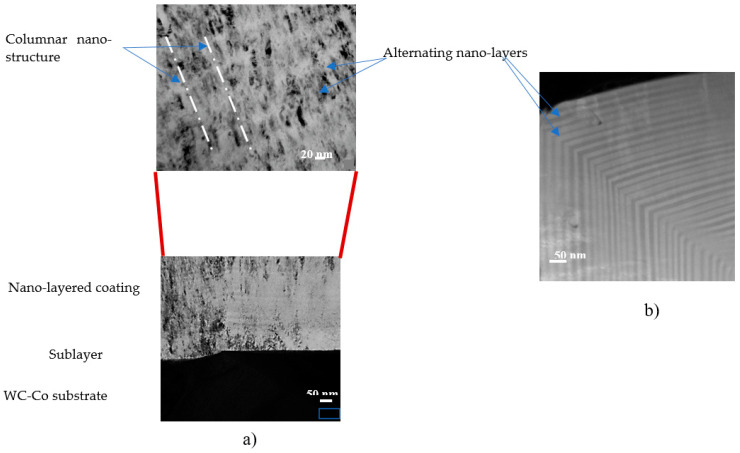
TEM (transmission electron microscope) image of FIB cross-section of TiAlCrSiYN/TiAlCrN bi-multilayer coating: (**a**) Thickness of alternating nano-layers in multilayered coatings is 20–40 nm, thickness of TiAlCrN sublayer is 100 nm; (**b**) FIB cross-section of the multilayer coating on the cutting edge of the tool [28].

**Table 1 nanomaterials-10-01720-t001:** Micro-mechanical properties of the studied coatings [28].

Coating	Thickness, Microns	Hardness, GPa	Reduced Elastic Modulus, GPa	H/E_r_ Ratio	H^3^/E_r_^2^ Ratio	CPRs Parameter L_c1_(L_c2_−L_c1_) ‘Scratch Crack Propagation Resistance’	Residual Stresses, GPa
Ti_0.2_Al_0.55_Cr_0.2_Si_0.03_Y_0.02_N/Ti_0.25_Al_0.65_Cr_0.1_N Multilayer	2	28.4 ± 4.5	327.1	0.087	0.212	1.9	−7.09 ± 0.6
Ti_0.2_Al_0.55_Cr_0.2_Si_0.03_Y_0.02_N/Ti_0.25_Al_0.65_Cr_0.1_N Bi-Multilayer	2	31.59 ± 2.5	329.07	0.095	0.291	5.8	−6.99 ± 0.5

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
