# Peer review of "Thin-Film PVD Coating Metamaterials Exhibiting Similarities to Natural Processes under Extreme Tribological Conditions"

_nanomaterials, 2020, doi:10.3390/nano10091720_

Round 1

Reviewer 1 Report

If the question is asked then please try to answer (or/and to give more information) in the paper.

  1. The English language needs some improvement. Examples: Line 193 “obstacle to aachieving”; Line 242 “effect.,”
  2. The paper is about TiAlCrSiYN/TiAlCrN coating. Probably it is reasonable to specify it in the title.
  3. The paper looks like Review of papers written previously by same authors. There is no much discussion of possible/available different approaches to obtain meta-materials or different meta-materials. Usually review paper contain analysis of information in form of tables…so reader can compare approaches and understand why one is better than another. Reference [56] or [75] exactly describes TiAlCrSiYN-based coatings. Probably it is reasonable to include comparison of different approaches or materials into paper if the authors wish to keep the same title.
  4. Line 18 “layers: an underlying layer of micro-scale (2-3 microns) thin-film PVD coatings”. Please rewrite since you use words “layer”, “film”, “coating” that has similar meaning. Similarly you use “micro-scale”, “2-3 microns”, “thin”.
  5. Line 31 Abstract should be concise. No need to repeat again “several micron 30 thick coating layer”.
  6. Please add references for all figures in their captions if they are taken from other sources.
  7. Figure 2. Please remove text in the upper part of the image since it is not relevant. If possible please improve the quality of image, place correctly the lines. If you show eV for some phases, then, please, show it for all others. Please indicate in the captions how long the given sample was tested (length of cut).
  8. Line 145-151 Please make it more clear and explain why sapphire is major tribo-phase and is it also primary? If it is also primary, then please just write in Line 152 “major primary tribo-phase (sapphire)”.
  9. Figure 3. According to Figure 8, cutting speed is 200-500 m/min (3-8 m/s) that enables to calculate that lifetime of tested cutting tool is about 1 minute. Is it correct? What is lifetime of usual tool materials in similar conditions?
  10. Figure 3. The “stable stage of wear” mentioned in captions is not shown on the curve and not mentioned in the text of paper.
  11. Figure 3. Please use the same X axis units for a, b and additional b image. Currently it is “m” or “s”. Why initial spike takes place when it is already 11.03 s of cutting?
  12. Figure 3. What is difference between indicated “Cutting forces” and “Resultant cutting force”?
  13. Figure 4. On Line 144 You mentioned that formation of “Ti, Al, Cr, Si, Y tribo-oxides” is possible. Why % of Al indicates “amount” of “thermal barrier films in total”?
  14. Line 236 “around 28-28 at. %”
  15. Figure 5. Left image: please indicate the position of the PVD coating. Please improve visibility of “Fe”.; Right image: please describe it in the figure captions.
  16. Line 263 “It is shown in Figure 1 that separate islands of non-equilibrium nitrides and oxides are formed”. It is hard to find any mentioning of the nitrides in Figure 1.
  17. Figure 6. Please use similar font size for each image. The symbol for Angstrom has dot on top of A “Å”. Or mention this in the captions.
  18. Line 281 “Wear velocity” May be wear rate?
  19. Line 286 “At a small (less than 1 m) beam size” What kind of beam?
  20. Line 312 “nano-scale t tribo-films layer” What is “t”? Probable word “film” or “layer” can be omitted.
  21. Line 314 “a very strong temperature gradient was achieved under extreme operational conditions, significantly reducing the temperature beneath the thermal barrier/lubricating nano-layer of the tribo-films as was clearly shown by TEM studies of 316 the worn coating layer”. Usually TEM alone is not enough to investigate temperature during tribological process. You need to mention that your study was combination of modelling and TEM.
  22. Line 326 “various tribo-films … performing a specific task” It is important to mention how they are arranged (one upon another, etc). How they all grow? May be scheme can be provided to explain it clearly.
  23. Figure 7c is not mentioned in caption.
  24. Line 350. “The functionally graded material replaces the sharp coating/substrate interface with a gradient … (Table 1)”. It is hard to find any information about “gradient interface” in Table 1.
  25. Table 1. Please mention the units for H/E and CPR.
  26. Line 354-377 Please remove repetition “One unique characteristic… the wear rate actually reduces.”
  27. Figure 8. Please reduce the size of your bars and show the upper limit of 500 deviation.
  28. Figure 9. (a) the scale bar is not well visible. It is not clear where is exactly the sublayer and nano-layers. Some layer are visible in the magnified image but the period is not 20-40 nm while rather close to 2 nm. It is not clear where is “modularly composed nano-multilayer structure (nano-layer period of 20-40 nm) with a columnar nano-structure”. Please add the scheme to explain the structure. (b) please add the scale bar. It is hard to conclude that (a) and (b) is the same coating.
  29. Please check the style of references.
  30. Reference 50 “…Machines (1970, Kiev, Ukraine: Techika.” Usually should be two brackets. Probably “Technika”.

Author Response

1. The English language needs some improvement. Examples: Line 193 “obstacle to aachieving”; Line 242 “effect.,”

Response: Correction is  made according to reviewer comment.

2. The paper is about TiAlCrSiYN/TiAlCrN coating. Probably it is reasonable to specify it in the title.

Response: in line 89-91 it was outlined that different approaches to the coatings design were used in surface engineering practice. Our analysis was focussed on the coatings with emergent performance (line 92).  Therefor the title was corrected in more generic way.

3. The paper looks like Review of papers written previously by same authors. There is no much discussion of possible/available different approaches to obtain meta-materials or different meta-materials. Usually review paper contain analysis of information in form of tables…so reader can compare approaches and understand why one is better than another. Reference [56] or [75] exactly describes TiAlCrSiYN-based coatings. Probably it is reasonable to include comparison of different approaches or materials into paper if the authors wish to keep the same title.

Correction is made according to reviewer comment. See lines 408-414.

4. Line 18 “layers: an underlying layer of micro-scale (2-3 microns) thin-film PVD coatings”. Please rewrite since you use words “layer”, “film”, “coating” that has similar meaning. Similarly you use “micro-scale”, “2-3 microns”, “thin”.

Correction is made according to reviewer comment. See lines 19-23.

5. Line 31 Abstract should be concise. No need to repeat again “several micron 30 thick coating layer”.

Correction is made according to reviewer comment. See lines 19-23.

6. Please add references for all figures in their captions if they are taken from other sources.

Correction is made according to reviewer comment.

7. Figure 2. Please remove text in the upper part of the image since it is not relevant. If possible please improve the quality of image, place correctly the lines. If you show eV for some phases, then, please, show it for all others. Please indicate in the captions how long the given sample was tested (length of cut).

Correction is made according to reviewer comment.

8. Line 145-151 Please make it more clear and explain why sapphire is major tribo-phase and is it also primary? If it is also primary, then please just write in Line 152 “major primary tribo-phase (sapphire)”.

Correction is made according to reviewer comment.

9. Figure 3. According to Figure 8, cutting speed is 200-500 m/min (3-8 m/s) that enables to calculate that lifetime of tested cutting tool is about 1 minute. Is it correct? What is lifetime of usual tool materials in similar conditions?

Response to reviewer. 500m/min is the surface speed of the tool and not relevant to the cut time. Calculations based on all cutting parameters used in the experiment show that 350 m length of cut corresponds to around 23 min of cutting. Life time of usual tool materials is incomparable. Carbide tools could be used only at the speed of around order of magnitude lower. Based on literature data similar coatings have only half of the tool life as compared to the presented values.

10. Figure 3. The “stable stage of wear” mentioned in captions is not shown on the curve and not mentioned in the text of paper.

Correction is made according to reviewer comment in figure 3 caption.

11. Figure 3. Please use the same X axis units for a, b and additional b image. Currently it is “m” or “s”. Why initial spike takes place when it is already 11.03 s of cutting?

Correction is made according to reviewer comment in figure 3 caption

12. Figure 3. What is difference between indicated “Cutting forces” and “Resultant cutting force”?

Response to reviewer. Meaning is the same.

13. Figure 4. On Line 144 You mentioned that formation of “Ti, Al, Cr, Si, Y tribo-oxides” is possible. Why % of Al indicates “amount” of “thermal barrier films in total”?

Response to reviewer. Because it is major element that is responsible for formation of all thermal barrier films: sapphire, mullite, garnet.

14. Line 236 “around 28-28 at. %”

Correction is made according to reviewer comment Figure 5. Left image: please indicate the position of the PVD coating. Please improve visibility of “Fe”.; Right image: please describe it in the figure captions.

16. Line 263 “It is shown in Figure 1 that separate islands of non-equilibrium nitrides and oxides are formed”. It is hard to find any mentioning of the nitrides in Figure 1.

Response to reviewer: Green islands are the nitrides as it is shown in the figure.

17. Figure 6. Please use similar font size for each image. The symbol for Angstrom has dot on top of A “Å”. Or mention this in the captions.

Correction is made in Figure caption

18. Line 281 “Wear velocity” May be wear rate?

Correction is made according to the reviewer comment.

19. Line 286 “At a small (less than 1 m) beam size” What kind of beam?

Response to reviewer. It is electron beam. Correction is made

20. Line 312 “nano-scale t tribo-films layer” What is “t”? Probable word “film” or “layer” can be omitted.

Correction is made according to the reviewer comment.

21. Line 314 “a very strong temperature gradient was achieved under extreme operational conditions, significantly reducing the temperature beneath the thermal barrier/lubricating nano-layer of the tribo-films as was clearly shown by TEM studies of 316 the worn coating layer”. Usually TEM alone is not enough to investigate temperature during tribological process. You need to mention that your study was combination of modelling and TEM.

Response to reviewer. TEM studies were combined with XANES studies, as it is outlined in the text (see corrections). Modeling is of nano-scale effects in tribology is an extremely challenging task that could be a goal of a special project.

22. Line 326 “various tribo-films … performing a specific task” It is important to mention how they are arranged (one upon another, etc). How they all grow? May be scheme can be provided to explain it clearly.

Response to the reviewer. The arrangement of tribo-films are shown in Fig.1 Correction is made

23. Figure 7c is not mentioned in caption.

Figure is corrected.

24. Line 350. “The functionally graded material replaces the sharp coating/substrate interface with a gradient … (Table 1)”. It is hard to find any information about “gradient interface” in Table 1.

Correction is made in the text according to reviewer comment.

25. Table 1. Please mention the units for H/E and CPR.

Response to reviewer: these are the ratio (H/E) and calculated values (CPR). No units though.

26. Line 354-377 Please remove repetition “One unique characteristic… the wear rate actually reduces.”

Correction is  made according to reviewer comment.

27. Figure 8. Please reduce the size of your bars and show the upper limit of 500 deviation.

Correction is  made according to reviewer comment.

28. Figure 9. (a) the scale bar is not well visible. It is not clear where is exactly the sublayer and nano-layers. Some layer are visible in the magnified image but the period is not 20-40 nm while rather close to 2 nm. It is not clear where is “modularly composed nano-multilayer structure (nano-layer period of 20-40 nm) with a columnar nano-structure”. Please add the scheme to explain the structure. (b) please add the scale bar. It is hard to conclude that (a) and (b) is the same coating.

Correction is made according to reviewer comment

29. Please check the style of references.

30. Reference 50 “…Machines (1970, Kiev, Ukraine: Techika.” Usually should be two brackets. Probably “Technika”.

Correction is  made according to reviewer comment

Reviewer 2 Report

First of all, the sources of figures should be given in the proper way. In fact, only the source of Fig. 1 is given. The rest of the figures has no references to their origin. I hope, authors aware of the common permission's granting system and policy.

Secondly, I would recommend modifying the title since it does not represent the content correctly.

Author Response

Comments and Suggestions for Authors

First of all, the sources of figures should be given in the proper way. In fact, only the source of Fig. 1 is given. The rest of the figures has no references to their origin. I hope, authors aware of the common permission's granting system and policy.

Response. We contacting editorial office and solved the permissions problem.

Secondly, I would recommend modifying the title since it does not represent the content correctly.

Response. The title is corrected. We tried to relate the title and the content in the best possible way. New text is added to justify the title. (lines 408-414).

Round 2

Reviewer 1 Report

If the question is asked, then please try to answer (or/and to give more information) in the paper.

  1. Not addressed properly. The paper looks like Review of papers written previously by same authors. There is no much discussion of OTHER possible/available approaches or OTHER meta-materials. Reference [56] or [75] exactly describes TiAlCrSiYN-based coatings.
  2. Not addressed properly. Figure 2. Please remove text in the upper part of the image since it is not relevant. If possible please improve the quality of image, place correctly the lines. If you show eV for some phases, then, please, show it for all others. Some part of text is not visible.
  3. Not addressed properly. Figure 3. According to Figure 8, cutting speed is 200-500 m/min (3-8 m/s) that enables to calculate that lifetime of tested cutting tool is about 1 minute. Is it correct? What is lifetime of usual tool materials in similar conditions? New: If authors used results of only “end mills” testing, then it should be mentioned in the article body (not only in the captions of Figure 3).
  4. Not addressed properly. Figure 3. The “stable stage of wear” mentioned in captions is not shown on the curve and not mentioned in the text of paper. New: The text and line in Figure 3a and 3b is partially deleted. Please improve quality. Probably you provide Word document with your figure that is changed during processing.
  5. Not addressed properly. Figure 4. On Line 144 You mentioned that formation of “Ti, Al, Cr, Si, Y tribo-oxides” is possible. Why % of Al indicates “amount” of “thermal barrier films in total”? New: Sapphire contains aluminium and oxygen …so content of aluminium is not the same as TOTAL content of sapphire.
  6. Not addressed properly. Figure 5. Left image: please indicate the position of the PVD coating. Please improve visibility of “Fe”.; Right image: please describe it in the figure captions.
  7. Not addressed properly. Line 278 “It is shown in Figure 1 that separate islands of non-equilibrium nitrides and oxides are formed”. It is hard to find any mentioning of the nitrides in Figure 1. New: In the Figure 1 caption it is written: “(b) – red image tribofilms; light green image -coating layer”. Do you mean that green is INITIAL coating or it is NEW coating formed during tribo-chemical reaction?
  8. Not addressed properly. Line 302 “At a small (less than 1 m) beam size” New: Probably it should be “um” or “nm”, not “m”.
  9. Not addressed properly. Table 1. Please mention the units for CPR. New: according to your reference [28]: “Micro-scratch tests were performed to a peak load of 5 N using the NanoTest Scratching Module with a 25 um radius diamond probe” It should be N*(N-N)…so unit is probably N in power of 2. In the paper http://dx.doi.org/10.1016/j.surfcoat.2014.07.007 CPR was measured in mN and results are up to 350 units (in your case it is up to 5.8).
  10. Not addressed properly. Figure 9. (a) the scale bar is not clearly visible.

Author Response

1. Not addressed properly. The paper looks like Review of papers written previously by same authors. There is no much discussion of OTHER possible/available approaches or OTHER meta-materials. Reference [56] or [75] exactly describes TiAlCrSiYN-based coatings.

Response to reviewer comment.

We add two references: 82, 83 to address this comment. We also added some discussion in lines 408-415. The text provided we outlined the features of metamaterials in general and justified unique features of the presented metamaterials.

2. Not addressed properly. Figure 2. Please remove text in the upper part of the image since it is not relevant. If possible please improve the quality of image, place correctly the lines. If you show eV for some phases, then, please, show it for all others. Some part of text is not visible.

Correction is made according reviewer comment.

3. Not addressed properly. Figure 3. According to Figure 8, cutting speed is 200-500 m/min (3-8 m/s) that enables to calculate that lifetime of tested cutting tool is about 1 minute. Is it correct? What is lifetime of usual tool materials in similar conditions? New: If authors used results of only “end mills” testing, then it should be mentioned in the article body (not only in the captions of Figure 3).

Correction is made according to reviewer comment: see line 80. We explained in our previous responses that calculation of life time provided by experts in around 32 min.

In our previous response lifetime of usual materials is indicated as well.

4. Not addressed properly. Figure 3. The “stable stage of wear” mentioned in captions is not shown on the curve and not mentioned in the text of paper.

Response. I do not understand this comment. Everything is already corrected in the figure and in the caption.

New: The text and line in Figure 3a and 3b is partially deleted. Please improve quality. Probably you provide Word document with your figure that is changed during processing.

Response. There is no deletion in Figures a and b. Cutting force data is presented vs. length of cut to address your previous comments.

5. Not addressed properly. Figure 4. On Line 144 You mentioned that formation of “Ti, Al, Cr, Si, Y tribo-oxides” is possible. Why % of Al indicates “amount” of “thermal barrier films in total”? New: Sapphire contains aluminium and oxygen …so content of aluminium is not the same as TOTAL content of sapphire.

Response to reviewer. As we explained previously thermal barrier tribo-films are Al-based. The only way to evaluate relative protective function of the surface is to evaluate relative Al content in the layer of tribo-films and coating. This data was published in many journals and no question aroused.

6. Not addressed properly. Figure 5. Left image: please indicate the position of the PVD coating. Please improve visibility of “Fe”.; Right image: please describe it in the figure captions.

Correction is made according to reviewer comment.

7. Not addressed properly. Line 278 “It is shown in Figure 1 that separate islands of non-equilibrium nitrides and oxides are formed”. It is hard to find any mentioning of the nitrides in Figure 1.

Response to reviewer.  Spots of localized destruction of ceramic tribofilms is the nitride coating layer.

New: In the Figure 1 caption it is written: “(b) – red image tribofilms; light green image -coating layer”. Do you mean that green is INITIAL coating or it is NEW coating formed during tribo-chemical reaction?

Green is initial coating. Coating is deposed once and it is wearing during cutting. No formation of coating during wear is taking place.

8. Not addressed properly. Line 302 “At a small (less than 1 m) beam size” New: Probably it should be “um” or “nm”, not “m”.

Correction is made according to the comment.

9. Not addressed properly. Table 1. Please mention the units for CPR.

Response. There are no units for CPR according to Ref 28. This is a relative value.

New: according to your reference [28]: “Micro-scratch tests were performed to a peak load of 5 N using the NanoTest Scratching Module with a 25 um radius diamond probe” It should be N*(N-N)…so unit is probably N in power of 2. In the paper http://dx.doi.org/10.1016/j.surfcoat.2014.07.007 CPR was measured in mN and results are up to 350 units (in your case it is up to 5.8).

Response. We are reporting our published data. Paper you mention measures this parameter using different loads. These data look incomparable.  

9. Not addressed properly. Figure 9. (a) the scale bar is not clearly visible.

Correction is made according to the comment.